# Spinal Cord Stimulation for Neuropathic Pain: Current Trends and Future Applications

**DOI:** 10.3390/brainsci8080138

**Published:** 2018-07-24

**Authors:** Ivano Dones, Vincenzo Levi

**Affiliations:** Neurosurgery Department, Functional Neurosurgery, Unit for the Surgical Treatment of Pain and Spasticity, Fondazione Istituto Neurologico Carlo Besta, 20131 Milan, Italy; vincenzo.levi@unimi.it

**Keywords:** spinal cord stimulation (SCS), neuromodulation, neuropathic pain

## Abstract

The origin and the neural pathways involved in chronic neuropathic pain are still not extensively understood. For this reason, despite the wide variety of pain medications available on the market, neuropathic pain is challenging to treat. The present therapeutic alternative considered as the gold standard for many kinds of chronic neuropathic pain is epidural spinal cord stimulation (SCS). Despite its proved efficacy, the favourable cost-effectiveness when compared to the long-term use of poorly effective drugs and the expanding array of indications and technical improvements, SCS is still worldwide largely neglected by general practitioners, neurologists, neurosurgeons and pain therapists, often bringing to a large delay in considering as a therapeutic option for patients affected by neuropathic chronic pain. The present state of the art of SCS in the treatment of chronic neuropathic pain is here overviewed and speculations on whether to use a trial period or direct implant, to choose between percutaneous leads or paddle electrodes and on the pros and cons of the different patterns of stimulation presently available on the market (tonic stim, high-frequency stim and burst stim) are described.

## 1. Introduction

The International Association for the Study of Pain defines neuropathic pain as the pain caused by a lesion or disorder of the somatosensory nervous system. It affects 7–10% of the general population, entailing an overall physical and psychological burden more relevant than that seen with nociceptive pain [1]. Although given the continuous development of new molecules appearing on the market to control neuropathic pain, this invalidating symptom is currently poorly improved by available drug treatments. In addition, common analgesic and opioid therapies carry a non-negligible risk of adverse events in the long term. Alternative options were lesional surgery at the dorsal root entry zone and, more recently, a number of neuromodulation procedures [2]. Among them, spinal cord stimulation (SCS) or dorsal column stimulation constitutes an advanced neuromodulation procedure able to actually decrease neuropathic pain in many syndromes such as in failed back surgery syndrome (FBSS), complex regional pain syndrome (CRPS) type I and II, postherpetic neuralgia and pure radicular pain [3]. Despite its proved efficacy, the favourable cost-effectiveness, when compared to the long-term use of poorly effective drugs and the expanding array of indications and technical improvements, spinal cord stimulation (SCS) is still worldwide largely neglected by general practitioners, neurologists, neurosurgeons and pain therapists, often bringing to a large delay in considering as a therapeutic option for patients affected by neuropathic chronic pain [4,5].

The present state of the art of SCS in the treatment of chronic neuropathic pain is overviewed and speculations on whether to use a trial period or direct implant, to choose between percutaneous leads or paddle electrodes and on the pros and cons of the different patterns of stimulation presently available on the market (tonic stim, high-frequency stim and burst stim) are described. 

## 2. Mechanism of Action

SCS owes its inception to the gate control theory (GCT), theorized by Wall and Melzack in their seminal 1965 paper. Wall and Melzack speculated that the nociceptive signal would be inhibited by antidromic activation of collateral, large, myelinated Aß fibres in the dorsal columns [6]. The first reported clinical application of dorsal column stimulation came 2 years later and at the time SCS was thought to act merely at the spinal segmental level [7]. However, The GCT theory did not take into account two evident SCS contradictions. The first is that, accordingly to the theory, SCS should be more effective in controlling acute nociceptive pain, which in fact is not the case. Secondly, Wall and Melzack’s theory is not able to explain the pain-free interval that is often noticed after discontinuation of stimulation [8]. For these reasons, the GCT theory seems to get more and more inconsistent to explain the mechanism of action of spinal cord stimulation in favour of other hypotheses, some of which involve the supraspinal pathway of pain control and transmission [9]. A pain-modulating dorsal column–brainstem–spinal loop was recently identified in animal models, while neuroimaging studies demonstrate that tonic SCS mainly acts by modulating the lateral pain ascending pathway and by interfering with the electrical and metabolic activity of the cingulate gyrus, lateral sensory thalamic nuclei, prefrontal cortex and postcentral gyrus [10,11]. Sato and colleagues showed that analgesic properties of SCS could be hampered by the use of opioid antagonists, thus suggesting that SCS might be also effective through the activation of the descending opioid pathway [12].

Several other experimental studies have elucidated the role of different transmitter systems which would be enhanced or inhibited by tonic dorsal column stimulation. SCS is deemed to neutralize the overexcitability of wide dynamic range (WDR) neurons in the dorsal horn by increasing γ-amino-butyric acid (GABA) release [13]. WDR neuron wind-up caused by excessive nociceptive inputs is believed to trigger the lateral pain pathway, giving the start to the abnormal transmission of pain sensation to the brain. So far, it remains unclear whether the SCS rebalancing effect of the system occurs solely as a result of presynaptic inhibition of the WDR neurons via antidromic activation or if it is due to most complex combined pre/postsynaptic phenomena [14]. Finally, evidence in experimental models in rats also suggests a role of the cholinergic transmitter systems. Increase in acetylcholine release was noticed under SCS even in association to the activation of the M4 muscarinic receptors, while low doses of muscarinic receptor agonist led to enhance the SCS-induced analgesic effect in rats [15,16].

However, the exact mechanism that allows the improvement of neuropathic pain observed in a large percentage of patients submitted to SCS still remains so far unclear.

## 3. SCS Indications and Patient Selection

Over the last years, a growing number of chronic pain syndromes of neuropathic origin have been treated with SCS, from brachial plexus and peripheral nerve injuries to postherpetic neuralgia and central pain of spinal cord origin, with varying grade of evidences [17] (Table 1).

To date, however, there are only two clinical pain syndromes that clearly benefit from SCS treatment: the failed back surgery syndrome (FBSS) and type 1 and 2 of the complex regional pain syndrome (CRPS). In Europe as well as in the United States, FBSS represents the most common indication for an SCS implant. Patients affected by FBSS are those who did not achieve satisfying outcome after single or multiple spinal operations in terms of pain relief, or who developed new, recurrent, drug-resistant low back or radicular pain regardless of the surgical procedure and possible surgical malpractice. This condition is often underrated, thus largely procrastinating the possible implant of SCS in favour of repetitive surgical procedures on the patient’s spine. A rate of recurrent back or leg pain of 5–36% in patients who had lumbar disc herniation surgery at a 2-year follow-up was recently reported in literature, whereas a prospective study by Skolasky et al. involving 260 patients who underwent surgical laminectomy with or without fusion for lumbar spinal stenosis secondary to degenerative alterations showed that 29.2% of patients had either no change or, even, increased pain at the 12-month follow-up after surgery [18,19]. Although there are only a few high-quality, large prospective and randomized comparative trials reported, the literature on SCS reports a large number of case series but only evidence supporting the use of SCS for the treatment of FBSS are of real significance. A systematic and comprehensive review regarding the effectiveness of SCS in treating chronic spinal pain showed that there is a clear (Level I–II) role for conventional low-frequency SCS as a treatment for otherwise intractable lumbar FBSS [20]. In another recent and extensive meta-analysis about conventional SCS for chronic back and leg pain, more than half of the patients experienced remarkable pain relief, independently from previous spinal surgery the patients possibly underwent. The pain remission was maintained during a mean follow-up period of 24 months [21]. In an exhaustive and thorough literature review, Cameron found an overall success rate of 62% among the 747 patients affected by FBSS and treated with SCS [22].

Although of minor incidence, particularly if compared to FBSS, the treatment of CRPS by SCS is also well established and includes one randomized controlled trial (RCT), which compared in a cohort of 54 patients SCS plus physical therapy with physical therapy alone [23]. At 6 months, in the SCS group pain was reduced by 3.6 on the visual analogue scale (VAS), while in the group receiving physical therapy alone, VAS was increased by 0.2 (*p* < 0.001). No clinically relevant improvement in functional status after 6-months follow-up was detected. The health-related quality of life improved only in 24 patients who underwent implantation of a spinal cord stimulator. Recently, the randomized prospective ACCURATE trial has compared SCS with another promising neuromodulation technique, the Dorsal Root Ganglion Stimulation (DRGS), which involves the percutaneous placement of a lead in the epidural posterosuperior space of the intervertebral foramen [24]. Both methods have been proved effective, but a higher statistical significance was associated with DRGS when considering pain relief, postural stability and mood improvement. Though efficacy of DRGS for CRPS treatment seems favorable, this surgical option is still in its inception. In addition, the data from the ACCURATE trial still need to be replicated, whereas there is more high-quality evidence to support the use of SCS [25].

Given the multitude of growing indications, appropriate patient selection is of paramount importance to achieve best SCS efficacy.

Patients who underwent surgical spine procedures may still suffer from unrecognized persistent compression of the neural elements. For this reason, a pre-operative spinal magnetic resonance imaging (MRI) should routinely be performed to search for an organic substrate of the pain. In that case, the patient should be considered for reoperation, otherwise SCS may be proposed as the next therapeutic option. Distinguishing neuropathic pain from other causes of pain may also be challenging. Over recent years, several helpful screening tools for the correct diagnosis of neuropathic pain have been validated. Among them, the Neuropathic Pain Questionnaire (NPQ), ID Pain and PainDETECT are widely available and easily deliverable, relying on interview questions only [26,27,28,29]. Finally, many guidelines have claimed the importance of a pre-operative psychological evaluation. This step may be precious for two reasons. Firstly, it greatly helps in excluding patients in whom a coexistence of major psychiatric diseases such as major depression, psychosis or drug abuse may hamper their response to stimulation. It must be said, however, that whether some studies found a negative association between depression and response to SCS, others did not, the evidence in literature thus being quite discordant [30,31,32]. 

In the case of appropriate indication and experienced implanter, SCS success rates are generally remarkable (in the range of about 50–75%). Despite all the careful selection pearls mentioned above, however, a variable percentage of patients do not benefit from SCS independently from the appropriateness of the indication. The cause of that partial response is still unclear; as also unclear is the average decrease in pain of 50% in the responders. Therefore, in order to increase the success rate of the procedure, a two-step surgery consisting of a trial stimulation phase before definitive internal pulse generator (IPG) implantation has become standard practice in most centres since the first SCS introduction. Kumar et al. found that about 17–20% of the patients decide not to proceed with the implantation although the trial period induced complete paraesthesia coverage of the painful region [33]. If the trial stimulation period has the indisputable advantage of avoiding probable unsuccessful implantation, on the other hand it carries a non-negligible risk of infection, which is reported to be between 2.4% and 18.6%, with consequent need of hardware removal and antibiotic therapy [34,35]. This wide range of infection rate reported is explained taking into account different factors, such as the single-centre surgical volume and surgeon experience [36]. Besides the risk of infection, the real utility and predictive value of the trial phase compared to direct permanent implantation has never been established through prospective, randomized, controlled trials. To date, the only paper addressing the topic is an Italian multicentre study enrolling 122 patients. In this paper, the authors assessed long-term clinical SCS efficacy in patients who were submitted to a trial period and in patients who, on the contrary, underwent immediate permanent implant. Significant reduction in pain, as measured by variation in visual analogue scale (VAS) score, was observed at least 1 year after implantation in both groups. Surprisingly, SCS efficacy was greater in patients who underwent permanent implant at once (59.5% vs. 71.4%). This difference, however, was not statistically relevant [37].

## 4. Technical Nuances

Despite the overall mini-invasive nature and straightforwardness of the procedure in experienced hands, several points regarding SCS surgical technique are still not standardized worldwide and deserve further discussion. One of these pertains to the choice between general and local anaesthesia. Many centres worldwide usually prefer to have the patient under local anaesthesia, using percutaneous-type electrode with the aid of fluoroscopic guidance. This strategy carries some clear advantages, the more obvious of which is to prevent any possible complication due to general anaesthesia and open surgery. In addition, many SCS experts claim that a percutaneous surgery in the awake patient is recommendable as it gives the opportunity to test the patient response to stimulation through his direct confirmation of full-paraesthesia coverage, thus confirming the correct electrode positioning [17]. Awake lead placement, however, has some shortcomings. It is well known that the success of any kind of awake surgery largely depends on the patient’s collaboration. Several individual patient’s factors, such as anxiety, stress or discomfort, should be taken into account and carefully assessed during the pre-operative screening before proceeding to awake surgery. Hence, a not-negligible portion of patients might not be able to tolerate the procedure. Moreover, awake surgery usually allows percutaneous leads to be easily positioned, whereas paddle lead requires a more invasive laminectomy approach in many cases. On the contrary, a lead electrode can move and get dislodged with body movements. In addition, it is well known from the literature that although the lead electrode is properly positioned, there may be a slight-to-moderate loss of SCS efficacy over time (in the range of 25–50%) due to both minor dislodgements of the electrode and the formation of scar tissue all around the leads [31,32]. Under these circumstances, a paddle lead might be more useful than a percutaneous one being steadily in contact with the dural surface and with negligible tilting or dislodgement in the long term. Moreover, a lead electrode produces a spherical electric field of which only the part toward the dural surface is effective. Conversely, a paddle electrode’s electric field is oriented to the spinal cord only, thus it needs less electric power to obtain similar results. Furthermore, when there is a need to have a widespread laterally extended electric field, one single paddle electrode with multiple lines of contacts can be used to shape the proper electric field that, on the contrary, with a lead electrode could be obtained only by positioning two separate electrodes. The Neurostimulation Appropriateness Consensus Committee (NACC) guidelines recently stated that “Confirmation of correct lead placement has been advocated with either awake intraoperative confirmation of paraesthesia coverage or use of neuromonitoring in asleep placement, such as Electromyography (EMG) responses or Somatosensory evoked potential (SSEP) collision testing.” [38]. To date, there is only a single prospective, multicentre study comparing safety and efficacy of the neuromonitoring-assisted asleep SCS implantation technique as compared to conscious procedures [39]. The authors found that SCS placement under general anaesthesia was a shorter procedure with superior paraesthesia coverage profiles, while maintaining lower adverse events and equal clinical outcomes for pain relief compared to awake surgery.

SCS is usually regarded as a safe procedure due to its reversible and minimally invasive characteristics [40]. Severe adverse events, such as spinal epidural bleeding and permanent neurologic deficit, are rare, whereas hardware complication and infection has been reported with an incidence of 24–50% and 7.5%, respectively [41,42,43] (Table 2).

It has been recently pointed out that the type of lead used may have an impact on both hardware complication and infection rate. In a single-centre prospective nonrandomized trial, Kinfe et al. compared effectiveness and safety of both lead types in a cohort of 100 patients who underwent SCS for FBSS with a 2-year follow-up [44]. They found a comparable clinical efficacy, but higher dislocation and infection rates in the group with cylindrical electrodes (14% and 10%, respectively) than in the group with paddle electrodes (6% and 2%, respectively). Another paper comprehensively analysed a large, independent, cohort of 131,774 patients from the United States who underwent percutaneous or paddle lead SCS placement comparing the incidence of complications, reoperation rates, and medication health-care costs both for percutaneous and paddle lead [45]. Placement of paddle leads was associated to a slightly higher initial postoperative complications, but with a significantly lower long-term reoperation rates. On the contrary, no difference in terms of health-care costs was noticed. Finally, in a retrospective study in a large cohort of 8326 patients conducted by Petraglia and colleagues [46], no significant difference in the rates of spinal cord trauma or spinal hematoma was observed between the two types of lead. In conclusion, current available data indicate an overall comparable and acceptable clinical efficacy and safety for both percutaneous and paddle lead. At the moment, the choice between the two types seems to mainly rely on individual implanter preference and background, while the choice between temporary implant and definite implant seems irrelevant in terms of percentage of good results in the long term, provided that an accurate selection of patients has been done.

## 5. Current and Future Development

The last decade saw an exponential technological advancement in the whole field of neuromodulation. Particularly, SCS therapy took advantage of the introduction of rechargeable generators, multiple leads paddle electrodes, position-sensing stimulation and MRI compatible devices that represent well-established great innovations [5]. Recently, this continuous innovative trend brought to an ongoing revolution on new different patterns of electric stimulation. Conventional SCS is based on a tonic pulse, released at constant frequency, (40–80 Hz), and a fixed pulse width of 200–450 μs and varying current amplitudes tailored on any patient needs [47,48]. This modality of stimulation is effective, but with a variable percentage of success that hardly is higher than 50% of pain control (in some series, a 50% pain relief is reported in approximately 50% of patients) and with a frequent progressive tolerance in the long term [21]. Consequently, there is an increasing need for new stimulation patterns, aimed both at improving SCS results in non-responders and avoiding long-term adaptation to the electrical therapy. In this respect, several types of new electric parameters are currently extensively investigated. Burst stimulation and high-frequency stimulation are the two main new stimulation options available so far. De Ridder et al. published a cohort of 12 patients who underwent the so-called “burst stimulation” [49]. This new stimulation pattern consists of intermittent trains of five high-frequency stimuli delivered at 500 Hz, 40 times per second and with a long pulse width and an interspike interval of 1000 μs delivered in constant-current mode. The monophasic pulses are charge-balanced at the end of the burst, differentiating it from clustered high-frequency tonic firing [50]. Applying this stimulation pattern, De Ridder and colleagues found that, when compared to conventional SCS, burst stimulation gave remarkable long-term pain higher suppression with a concomitant greater reduction in the number of patients sensing paraesthesias due to stimulation (92% vs. 17% of patient, respectively). In addition, a major extension of the stimulation effect to the midline region seems, unlike conventional tonic stimulation, to be observed during burst stimulation. This major advantage seems to be ascribed to the higher chance of intercepting even deep nerve fibres by means of trains of impulses at higher frequency [51].

Another paraesthesia-free technique is the high-frequency continuous stimulation. High-frequency stimulation is similar in principle to tonic stimulation, using 30 μs pulse width and individually actively charge-balanced pulses delivered at very high frequency (10 kHz) [47,52]. It is based on the staggered implantation of two 8-contact electrodes at the thoracic level (T8 down to T12) and although the reason for its inception is still unclear, it is thought to decrease WDR neuron firing rates and their consequent wind-up phenomenon [53]. At the moment, however, this assumption is not supported by any experimental or clinical evidence [8]. Regarding its clinical efficacy, in a recent prospective multicentre study 70% of patients treated by high-frequency SCS experienced a significant and sustained low back pain and leg pain relief, greater than 50%, without referring any concomitant induced paraesthesia [53]. On the other hand, no significant differences were found between a short trial period (2 weeks) of sham stimulation and high-frequency 5 kHz stimulation in a randomized study including 33 patients [54]. 

Given these considerations, it is not yet possible to draw final conclusions on the real pain-relieving efficiency of both burst and high-frequency stimulation. Although these new stimulation modalities seem very promising, further prospective randomized clinical trials are needed to prove their presumed clinical superiority over conventional tonic SCS.

## 6. Conclusions

Although still underused, conventional SCS may be considered as an effective, safe, well-tolerated and reversible treatment option for severe drug-refractory neuropathic pain. Accurate indications and cautious patient selection represent the principal mainstays for the success of this treatment. In the near future, there will surely be confirmations as to the efficacy of the new patterns of stimulation both at high frequency and through burst stimulation and, possibly, future new patterns to improve the efficacy of this treatment in improving chronic neuropathic pain. 

## Figures and Tables

**Table 1 brainsci-08-00138-t001:** Common SCS indications and contraindications.

SCS Common Neuropathic Indications	SCS Main Contraindications
Failed back surgery syndrome Complex regional pain syndrome (I and II) Radicular and nerve root pain Postherpetic neuralgia Pain due to peripheral nerve injury Intercostal neuralgia Phantom pain	Infection Coagulopathy Spinal stenosis Psychiatric disorders Substance abuse

SCS: spinal cord stimulation.

**Table 2 brainsci-08-00138-t002:** SCS surgical complications.

SCS Common Complications
More frequent Hardware-related (lead migration, breakage, connection failure, malfunctioning, pain at the IPG) Haematoma and seroma at IPG site
Rare Spinal epidural haematoma CSF leak Neurological deficit

IPG: internal pulse generator; CSF: cerebrospinal fluid.

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
