# Peer review of "Spinal Cord Stimulation for Neuropathic Pain: Current Trends and Future Applications"

_brainsci, 2018, doi:10.3390/brainsci8080138_

Reviewer 1 Report

Review of manuscript entitled “Type of manuscript: Review

Title: Spinal cord stimulation for neuropathic pain: current trends and future applications.

In this manuscript, the authors give a review of Spinal Cord Stimulation for neuropathic pain. The manuscript is written in a clear and straightforward manner. The literature is presented in a comprehensible way.

With regard to contents this review article the mechanisms of action and the indications and patient selection are described in a condensed way. The chapter termed “technical nuances” is slightly focused on the debate whether or not to use paddle leads as a first instance treatment. It becomes quite obvious that the authors` position in this discussion is in favour of paddle leads. In my view it is adequate to point out the personal view as long as all arguments pro and con paddle leads have been referred. I think that this has widely been accomplished in the manuscript, although I am personally more convinced of the use of percutaneous leads, as they are less invasive and do not require general anesthesia.  

I have only minor suggestions for improvement:

 P 1 L 1  Review (instead of “Revirw”)

P 1 L 11 “painkillers” may be change to “pain medication”

P1 L 18 There seems to be a change in the font size.

P3 L83 “To date, however, there are only two clinical syndromes”

Maybe insert “pain” after “clinical”

 P3 L101 ff

“In another recent and extensive meta-analysis about conventional SCS for chronic back and leg pain, more than half of the patients experienced remarkable pain relief, independently from previous spinal surgery the patients possibly underwent. The pain remission was maintained during a mean follow-up period of 24 months.[21]”

In this context is also the review by Cameron (2004), who found success rates ranging between 57 and 84%, should be quoted.

 Reviewer 2 Report

This is an important piece of work, which definitely contributes to a deepr insight of the actual technique and clinical potential of perpiheral neuromodulation. It is well written and interesting to read.

I do have however some minor comments to be improved prior to a final decision:

CRPS is not anymore considered a first line indication for SCS, especially since spinal ganglion (DRG) stimulation has been shown to have a much better effect, Basically DRG should anyway mentioned , explained and validated in the paper, since the technical approach is roughly similar to SCS and the indications are overlapping each other. Although surgical and also programming malpractice might induce a "deep learning curve", it's potential to improve clinical long-term results is evident. In order to find electrophysiological biomarkers for treatment success the recording of evoked compound action potentials ECAPs might be a valuable tool in the future, at least one device company is upcoming with auch a technique.

Finally it is important to mention ( as the authors basically did) that chronic opiods have a significantly less benefit for neuropathic pain as SCS/DRG and finally any medication has a much higher potential to harm those patients.